# Utilitarian Algorithm Configuration for Infinite Parameter Spaces

**Devon R. Graham & Kevin Leyton-Brown**
Department of Computer Science
University of British Columbia
Vancouver, BC
{drgraham,kevinlb}@cs.ubc.ca

## Abstract

Utilitarian algorithm configuration is a general-purpose technique for automatically searching the parameter space of a given algorithm to optimize its performance, as measured by a given utility function, on a given set of inputs. Recently introduced utilitarian configuration procedures offer optimality guarantees about the returned parameterization while provably adapting to the hardness of the underlying problem. However, the applicability of these approaches is severely limited by the fact that they only search a finite, relatively small set of parameters. They cannot effectively search the configuration space of algorithms with continuous or uncountable parameters. In this paper we introduce a new procedure, which we dub COUP (Continuous, Optimistic Utilitarian Procrastination). COUP is designed to search infinite parameter spaces efficiently to find good configurations quickly. Furthermore, COUP maintains the theoretical benefits of previous utilitarian configuration procedures when applied to finite parameter spaces but is significantly faster, both provably and experimentally.

## 1 Introduction

*Algorithm configuration* is a general-purpose technique for optimizing the performance of algorithms by automatically searching over the space of their input parameters. Given a set of possible parameter assignments (i.e., a set of "configurations") our objective is to find one that makes the algorithm perform well on average, over a set of inputs of interest. Recently, attention has begun to shift from procedures that define performance in terms of average runtime toward procedures that define performance in terms of more general utility functions over runtime. Graham et al. (2023b) introduced Utilitarian Procrastination (UP), the first non-trivial algorithm configuration procedure that maximizes utility achieved instead of minimizing runtime. Unlike more naive approaches, UP can adapt to the hardness of the problem it faces, requiring provably less time to guarantee optimality when many of the configurations it considers are suboptimal. However, UP requires that this set of candidate configurations be discrete and relatively small. This is a major limitation; e.g., many algorithms have at least one continuous parameter. UP cannot effectively search over continuous spaces.

In this paper we present COUP (Continuous, Optimistic Utilitarian Procrastination), which is designed to handle infinite (e.g., continuous) configuration spaces. Like UP, COUP is utilitarian and takes inspiration from the world of multi-armed bandits. However, COUP is based on a different bandit procedure that is guided by the principle of "optimism in the face of uncertainty" and is more readily adapted to handle large configuration spaces, including those with continuous parameters. COUP also offers significantly better performance than UP on finite sets of configurations because COUP does not try to explicitly rule bad configurations out, but instead simply ignores them once they no longer appear as promising as other alternative configurations.

Section 2 discusses relevant background material. Section 3 introduces the finite-configuration-space version of COUP and shows that this version finds an approximately optimal configuration in strictly less total time than UP. We simply call this version OUP since it, like UP, does not actually search over a continuous parameter space. Section 4 introduces COUP, leveraging our understanding of

the finite case into a general-purpose procedure. We discuss empirical evaluations in Section 5 and conclude with Section 6.

## 2 BACKGROUND

Given a parameterizable algorithm and a set of possible parameter assignments, the goal of *algorithm configuration* is to find a parameter assignment that makes the algorithm perform well across a set of given inputs. If we also require that a procedure gives a *guarantee* about the near-optimality of its returned configuration, algorithm configuration can be seen as a multi-armed bandit problem (see e.g., Lattimore & Szepesvári (2020)) called *approximate best-arm identification*: finding an arm whose mean reward is close to the reward of the best arm available. In this case, it is not the regret of the procedure that concerns us, but the number of samples needed to guarantee the near-optimality of the returned arm. Even-Dar et al. (2002) observe that if we want to find an arm whose value is within $\epsilon$ of the best with good probability, we could just take $\tilde{O}(\epsilon^{-2})$ samples of each arm and apply Hoeffding's inequality (Hoeffding, 1963) to obtain upper and lower confidence bounds on the mean reward of each arm that are sufficient to identify an approximately optimal one (i.e., one within an $\epsilon$ additive factor of optimal). However, this will be more samples than necessary in all but the worst case; any arm that is significantly worse than the best arm can be ruled out using fewer samples than are required by naively applying Hoeffding's inequality. The Successive Elimination (SE) procedure (Even-Dar et al., 2002) is able to identify an approximately optimal arm with only $\tilde{O}(\min\{\Delta_i^{-2}, \epsilon^{-2}\})$ samples of each arm, where $\Delta_i$ is the suboptimality gap for arm $i$. This can be a significant reduction when many arms are far from optimal. SE achieves this improved, *adaptive* bound by being *anytime*: it does not take an $\epsilon$ as input. It starts by taking only a few samples of each arm, guaranteeing a relatively large $\epsilon$, and then improves this over time. This allows it to stop sampling bad arms and rule them out before it is finished. Up to logarithmic factors, taking $m = \min\{\Delta_i^{-2}, \epsilon^{-2}\}$ samples of arm $i$ is not only sufficient, but also necessary for general inputs (Mannor & Tsitsiklis, 2004).

SE is a simple and intuitive procedure with approximately optimal sample complexity, but it can tend to over-investigate bad arms, sampling them repeatedly until it can expressly rule them out. The Upper Confidence Bound (UCB) procedure (Lai, 1987) is an alternative to SE that focuses on arms that look promising and/or about which little is known. By repeatedly pulling the arm with largest upper confidence bound, UCB attempts to maximize the information it gains with each pull. For more discussion on the difference between SE and UCB and on their relative performance see the survey by Jamieson & Nowak (2014).

There are two key differences between algorithm configuration and most other bandit problems. First, each arm pull imposes a stochastic cost arising from the amount of time it takes for the algorithm run to complete. Second, we have the option of bounding these costs by "capping" algorithm runs at any fixed time of our choice, but when we do so we obtain only censored information about the arm pull (we learn that the run would have taken more than the captime but not how much more). Overall, it is not the total number of samples that we want to control, but the total amount of time spent searching.

Most existing algorithm configuration procedures that offer theoretical guarantees aim to minimize average runtime. These can broadly be seen as implementing either SE or UCB. These procedures either run all existing configurations simultaneously (i.e., round-robin or in parallel), ruling out the ones that can be proved to be suboptimal after each step (Weisz et al., 2018; 2019; 2020; Brandt et al., 2023), or they iteratively select a particular configuration according to a confidence-bound index, and run only this configuration (Kleinberg et al., 2017; 2019). In either case, these procedures are significantly complicated by the fact that runtimes are potentially unbounded, and even the notion of optimality that they target requires adjustment to compensate for this. Other theoretically-motivated approaches offer performance guarantees based on measures of complexity and guarantee notions of PAC optimality akin to those we present here (Gupta & Roughgarden, 2017; Balcan et al., 2017; 2021). The focus there is not on the time consumed by the configuration process, but on the number of samples required to find a sufficiently good configuration.

*Utilitarian* algorithm configuration procedures do not try to minimize expected runtime, but instead try to maximize expected utility, as measured by a user-specified utility function that captures the value associated with solving an input instance as a (weakly decreasing, but not necessarily linear) function of the amount of time spent solving it. For example, if we face daily scheduling problems, then getting a solution after waiting 200 days is not 100 times worse than getting it after two days; both are too

long to wait. If one algorithm solves most inputs very quickly but occasionally takes an astronomical amount of time, while another algorithm takes a moderate amount of time on every input, determining which algorithm is "better" will likewise depend on our particular setting: How much time do we have? How much better is it to get an answer quickly? In Graham et al. (2023a) we generalized such intuitions, offering a detailed argument from first principles that utility maximization is a more appropriate objective (e.g., for algorithm configuration) than runtime minimization and showing that such utility functions are guaranteed to exist if our preferences for algorithm runtime distributions follow a certain reasonable structure based on the axioms of Von Neumann & Morgenstern (1947).

UP (Graham et al., 2023b) is a utilitarian configuration procedure that can be seen as SE coupled with a doubling mechanism for discovering sufficiently large captimes. UP has several desirable qualities that we would like to incorporate into our own configuration procedure. First, UP maximizes utility instead of minimizing runtime. Second, unlike many runtime-optimizing procedures UP can make simple theoretical guarantees about the near-optimality of the configuration it returns. Third, UP is *anytime*, meaning that it provides a better guarantee the longer it is run, and so requires minimal initial input from the user, who can simply observe the procedure's execution and terminate it when they are satisfied with the guarantee being made. Finally, UP's guaranteed performance is *adaptive* to the inputs it receives; it provably performs better on non-worst-case inputs (i.e., the associated theoretical guarantees are *input-dependent*).

However, UP can only be applied to a finite (and relatively small) set of configurations since it cycles through these one by one and runs each of them in turn. This is a prohibitive limitation we would like to overcome. A natural idea would be to sample a finite set from the infinite configuration space and simply run UP on this set. The optimality guarantee offered by UP would then be made with respect to the best configuration in this subset, and if enough configurations are sampled, we can guarantee that our subset will be likely to contain at least one of the top, say, $\gamma$-fraction of all configurations. This strategy will work, but since the choice of $\gamma$ potentially limits the quality of the configuration we are likely to find, we would like to make our procedure anytime in $\gamma$ as well; we would like our procedure to start with a relatively small set of configurations (i.e., a relatively large $\gamma$) and grow this set as time goes on, shrinking the $\gamma$ it can guarantee. Of course we still wish to be anytime with respect to the accuracy parameter $\epsilon$, and here we run into a problem with UP. Since UP loops over all remaining configurations, at any point during its execution it will have run all of these configurations on effectively the same number of instances (at most the difference is one). What should we do when we want to expand the set of configurations we are considering? Any newly added configuration will have a large upper confidence bound, since we will know little or nothing about it. This will severely limit the anytime $\epsilon$ we can guarantee because it remains possible that the new configuration is much better than all the existing ones. So new configurations would need to be "caught up" when they are added. How should we do this?

One extreme strategy is to completely "catch up" new configurations by running them on all existing instances. This is unnecessarily costly because it fails to leverage what has already been learned about promising configurations. At the other extreme, we could simply refrain from "catching up" new configurations at all, just adding them to the existing pool, forever estimating them with fewer samples and thus lower accuracy. This would effectively limit the $\epsilon$ we could guarantee to the lower accuracy of these newly introduced configurations. Is there a way to balance these two extremes and integrate new configurations into the estimation process smoothly? This leads us to the UCB procedure. In terms of the runtime cost we incur, we would prefer to take the second extreme approach above and not do anything special with new configurations. However, this would lead to configurations with relatively large upper confidence bounds, which would limit the optimality guarantee we could make. If we choose instead to focus on configurations that have large upper bounds, as UCB does, we will explore new configurations just up to the point where something else looks better, and then stop.

Thus, whereas UP was built around the SE procedure, COUP is built around the UCB procedure. Like UP, COUP is utilitarian, adaptive, and anytime with respect to the optimality parameter $\epsilon$. At the same time, COUP can also be applied to infinitely-large sets of configurations, and is also anytime with respect to $\gamma$, the fraction of unexplored configurations. COUP initially makes guarantees for a small set of configurations, then expands this set while simultaneously improving the optimality guarantee that it can make with respect to this (growing) set. What's more, COUP also beats UP at its own game: COUP is significantly faster than UP when the input actually is a fixed, finite set of configurations. Because it is built on the SE algorithm, UP runs all suboptimal configurations until they can explicitly be eliminated. For some sets of inputs this can be necessary but in many cases it is

more efficient to rule out poor-performing configurations by learning more about (i.e., by narrowing the confidence bounds of) other, better-performing configurations. COUP provably stops running sub-optimal configurations well before they would be eliminated by UP. Additionally, we re-evaluate a key component of UP—the condition it checks before doubling a configuration's captime—and propose an improvement. UP attempts to balance the estimation error due to sampling with the error due to capping. We argue that this is better achieved with a different doubling condition that emerges from a more careful analysis of the confidence bounds.

## 3 THE CASE OF FEW CONFIGURATIONS: OUP

We begin by presenting and analyzing the finite-configuration-space version of COUP, which we call OUP (Algorithm 1) to indicate that it does not search over continuous spaces like COUP does. OUP can most easily be understood as a UCB procedure for finding configurations with good *capped* mean utility, along with a periodic doubling scheme that allows larger and larger captimes to be considered. Like UP, it is adaptive and anytime in the optimality parameter $\epsilon$.

We are given a set of configurations, indexed by $i$, and a set of instances, indexed by $j$. We will use $t_{ij}$ to denote the true uncapped runtime of configuration $i$ on input $j$, and $t_{ij}(\kappa) := \min(t_{ij}, \kappa)$ to be the $\kappa$-capped runtime of $i$ on $j$. When we do a run, we observe $t_{ij}(\kappa)$ rather than $t_{ij}$; these coincide for any run that completes. The instance distribution $\mathcal{D}_J$, along with any randomness of the algorithm or execution environment will together induce a runtime distribution for each configuration $i$. We will use $\mathcal{D}_i$ to denote this runtime distribution, and $F_i$ to denote its CDF. For each configuration $i$, the true uncapped expected utility is $U_i := \mathbb{E}_{t \sim \mathcal{D}_i}\big[u(t)\big]$. The cumulative distribution function is $F_i(\kappa) := \Pr_{t \sim \mathcal{D}_i}\big(t \leq \kappa\big)$. The capped expected utility is $U_i(\kappa) := \mathbb{E}_{t \sim D_i}\big[u\big(\min(t, \kappa)\big)\big]$. Finally, the optimality gaps $\Delta_i := \max_{i'} U_{i'} - U_i$ partially determine the hardness of the problem; if a configuration $i$ has large optimality gap $\Delta_i$, then it will be easier to rule out. The notion of optimality that we target is given by the following definition.

**Definition 1** ($\epsilon$-optimal). *A configuration $i$ is $\epsilon$-optimal if $\Delta_i \leq \epsilon$.*

The error associated with $m$ runtime samples at captime $\kappa$ is $\alpha(m, \kappa) := \sqrt{\frac{\ln(11nm^2(\log \kappa + 1)^2/\delta)}{2m}}$, chosen to satisfy Hoeffding's inequality and the necessary union bounds. An execution of OUP will be called *clean* if at all times during its execution we have $\big|\widehat{F}_i - F_i(\kappa_i)\big| \leq \alpha(m_i, \kappa_i)$ and $\big|\widehat{U}_i - U_i(\kappa_i)\big| \leq \big(1 - u(\kappa_i)\big) \cdot \alpha(m_i, \kappa_i)$ for all configurations $i$. So during a clean execution, empirical capped average utilities are close to true capped average utilities and empirical CDF values are close to true CDF values at the captime $\kappa_i$.

**Lemma 1.** *An execution of OUP is clean with probability at least $1 - \delta$.*

All proofs are deferred to the appendices. The idea behind Lemma 1 is that the bounds that define a clean run hold initially, they do not change in any round where $i$ is not selected, and by Hoeffding's inequality they hold with high probability in any round where $i$ is selected. The next lemma shows that during a clean execution, the confidence bounds will be valid, and also not too far apart.

**Lemma 2.** *For all $i$ at all points during a clean execution of OUP we have $LCB_i \leq U_i \leq UCB_i$ and $UCB_i - LCB_i \leq 2\alpha(m_i, \kappa_i) + u(\kappa_i)\big(1 - F_i(\kappa_i)\big)$.*

The upper and lower bounds are defined in Algorithm 1. Each iteration of OUP's main loop corresponds to an iteration of the UCB procedure, with an additional check to see if the captime needs to be doubled. In each iteration the configuration with largest upper bound is selected (Line 7). The doubling condition is then checked (Line 9) and, if necessary, instances that had previously capped are rerun at the newly doubled captime (Line 11). A new runtime sample is drawn, and the sample mean and confidence bounds are recomputed. Finally, we attempt to eliminate provably suboptimal configurations (Lines 20-24). This loop repeats until terminated by the user, at which point it returns the configuration with largest lower confidence bound, which is the configuration we can prove the smallest suboptimality for. We can now state the main theorem of this section.

**Theorem 1.** *With probability at least $1 - \delta$, OUP eventually returns an optimal configuration and it returns an $\epsilon$-optimal configuration if it is run until $2\alpha_{i^{opt}} + u(\kappa_{i^{opt}})\big(1 - F_{i^{opt}}(\kappa_{i^{opt}})\big) \leq \epsilon$. Furthermore, for any suboptimal configuration $i$, if $m_i$ and $\kappa_i$ ever become large enough that*

$$2\alpha(m_i, \kappa_i) + u(\kappa_i)\big(1 - F_i(\kappa_i)\big) < \Delta_i \tag{1}$$

---

**Algorithm 1** OUP

---

1: **Input:** configurations $i = 1, ..., n$; instances $j = 1, 2, ...$; utility function $u$; failure probability $\delta$.
2: $I \leftarrow \{1, ..., n\}$            ▷ candidate configurations
3: **for** $i \in I$ **do**
4:      $LCB_i \leftarrow 0; UCB_i \leftarrow 1; \widehat{U}_i \leftarrow 0; \widehat{F}_i \leftarrow 0; m_i \leftarrow 0; \kappa_i \leftarrow 1$      ▷ initializations
5: **end for**
6: **for** $r = 1, 2, 3, ....$ **do**
7:      $i \leftarrow \arg\max_{i' \in I} UCB_{i'}$
8:      $m_i \leftarrow m_i + 1$
9:      **if** $2\alpha(m_i, \kappa_i) \leq u(\kappa_i)(1 - \widehat{F}_i)$ **then**      ▷ captime doubling condition
10:          $\kappa_i \leftarrow 2\kappa_i$
11:          $t_{i1}(\kappa_i), ..., t_{im_i}(\kappa_i) \leftarrow$ runtime of configuration $i$ on instances $1, ..., m_i$ with timeout $\kappa_i$
12:      **else**
13:          $t_{im_i}(\kappa_i) \leftarrow$ runtime of configuration $i$ on instance $m_i$ with timeout $\kappa_i$
14:      **end if**
15:      $\widehat{F}_i \leftarrow \frac{|\{j \in [m_i] : t_{ij}(\kappa_i) < \kappa_i\}|}{m_i}$      ▷ fraction of runs that completed
16:      $\widehat{U}_i \leftarrow \frac{1}{m_i} \sum_{j=1}^{m_i} u(t_{ij}(\kappa_i))$      ▷ empirical average utility
17:      $UCB_i \leftarrow \widehat{U}_i + (1 - u(\kappa_i)) \cdot \alpha(m_i, \kappa_i)$
18:      $LCB_i \leftarrow \widehat{U}_i - \alpha(m_i, \kappa_i) - u(\kappa_i)(1 - \widehat{F}_i)$
19:      $i^* \leftarrow \arg\max_{i' \in I} LCB_{i'}$
20:      **for** $i' \in I$ **do**
21:          **if** $UCB_{i'} < LCB_{i^*}$ **then**      ▷ if $i'$ is suboptimal
22:              $I \leftarrow I \setminus \{i'\}$      ▷ remove $i'$ from consideration
23:          **end if**
24:      **end for**
25:      **if** execution is terminated or $|I| = 1$ **then**
26:          **return** $i^*$
27:      **end if**
28: **end for**

---

*then $i$ will never be run again, and $i$ will be outright eliminated once $m_i, \kappa_i, m_{i^{opt}}$ and $\kappa_{i^{opt}}$ are large enough that*

$$2\alpha(m_i, \kappa_i) + 2\alpha(m_{i^{opt}}, \kappa_{i^{opt}}) + u(\kappa_i)(1 - F_i(\kappa_i)) + u(\kappa_{i^{opt}})(1 - F_{i^{opt}}(\kappa_{i^{opt}})) < \Delta_i. \quad (2)$$

If OUP keeps selecting and running $i$, then the term $2\alpha(m_i, \kappa_i) + u(\kappa_i)(1 - F_i(\kappa_i))$ will continue to shrink. So we will eventually have $2\alpha(m_i, \kappa_i) + u(\kappa_i)(1 - F_i(\kappa_i)) < \Delta_i$ as long as we keep running $i$. In either case, we will eventually stop running any suboptimal $i$. This puts a hard limit on the amount of time we will ever dedicate to any suboptimal configuration, and this limit is better than the corresponding guarantee made by UP: compare our Theorem 1 to Theorem 4 in Graham et al. (2023b). UP guarantees that $i$ will be eliminated once the bound in Equation (2) is satisfied. The bound in Equation (1) guaranteed by OUP is a necessary condition for this. OUP does not keep running suboptimal configurations until they can be eliminated, as UP does. Instead, it simply stops running them, while continuing to tighten the bounds of better configurations instead.

Algorithm 1 is stated using the old doubling condition introduced in UP. We propose an alternative based on examining the confidence width $UCB_i - LCB_i = 2(1 - u(\kappa_i)) \cdot \alpha(m_i, \kappa_i) + u(\kappa_i)(1 - \widehat{F}_i + \alpha(m_i, \kappa_i))$. The first term is the error from runs below $\kappa_i$ and the second term is the error from runs above $\kappa_i$. Our new doubling condition attempts to balance these two terms. Due to space constraints we discuss this more in Appendix B, but ultimately Line 9 of Algorithm 1 becomes

$9'$: **if** $2(1 - u(\kappa_i)) \cdot \alpha(m_i, \kappa_i) \leq u(\kappa_i)(1 - \widehat{F}_i + \alpha(m_i, \kappa_i))$ **then**    ▷ captime doubling condition

The change to the continuous-space version COUP is analogous (Line 18 of Algorithm 2). Implementing this refined doubling condition improves the performance of both UP and (C)OUP.

## 4   THE CASE OF MANY CONFIGURATIONS: COUP

We now assume we have a very large, possibly uncountably infinite, set of configurations $\mathcal{A}$, along with an associated distribution $\mathcal{D}_{\mathcal{A}}$ over configurations $a \in \mathcal{A}$ from which we sample. It is no longer feasible to target the configuration with maximum utility $OPT = \max_{a \in \mathcal{A}} U_a$. It may be arbitrarily unlikely that we ever sample this configuration and, in fact, such a maximum may not even exist. So we must relax our objective. For any $\gamma \in (0,1)$, let $OPT^{\gamma} = \sup \left\{ \mu \ : \ \Pr_{a \sim \mathcal{D}_{\mathcal{A}}}[U_a \le \mu] \le 1 - \gamma \right\}$. The quantity $OPT^{\gamma}$ marks the top $\lfloor 1/\gamma \rfloor$-quantile of expected utilities. That is, $OPT^{\gamma}$ is the utility of the best configuration that remains after we have excluded the top $\gamma$-fraction of configurations. The notion of optimality that we will target is given by the following definition.

**Definition 2** (($\epsilon, \gamma$)-optimal). *A configuration $a \in \mathcal{A}$ is ($\epsilon, \gamma$)-optimal if $U_a \ge OPT^{\gamma} - \epsilon$.*

We present the general version of COUP in Algorithm 2. The procedure runs in successive phases. Each phase is an execution of the few-configuration version, OUP, with the bounds being chosen carefully so that runs can be shared across phases. Each phase $p$ of COUP is characterized by two parameters: $\gamma_p \in (0,1)$ determines the tail quantile being targeted in this phase, and $\epsilon_p \in (0,1)$ determines the level of optimality being targeted. At the start of each phase $p$, COUP samples enough new configurations to ensure that it has a total of $n_p = \frac{1}{\gamma_p} \ln \frac{\pi^2 p^2}{3\delta}$ configurations, where $\delta$ is the failure probability and $\pi$ is the mathematical constant (Lines 5 and 6). COUP then essentially runs the finite-space version OUP on this set of $n_p$ configurations until it can prove $\epsilon_p$-optimality (Lines 15–28). However, COUP "maintains state" between phases so that it can build on what it has already learned. Confidence bounds in phase $p$ incorporate any runs performed in previous phases (Lines 12 and 13), with the error terms being set carefully so that the total failure probability is controlled across all phases.

If $\gamma_p$ is relatively small and $\epsilon_p$ is relatively large, then COUP will sample a large number of configurations in phase $p$, but not work too hard to prove that any of them is approximately optimal. On the other hand, if $\gamma_p$ is relatively large and $\epsilon_p$ is relatively small, then COUP will sample only a few configurations and work very hard to prove one of them is nearly optimal (with respect to this set). Since we are interested in procedures that are anytime in both $\epsilon$ and $\gamma$, we will be interested in sequences of parameters for which $\epsilon_p \to 0$ and $\gamma_p \to 0$ as $p \to \infty$. We can think of $\epsilon_p$ as controlling the rate at which we explore new instances, and $\gamma_p$ as controlling the rate at which we explore new configurations. They are dissimilar in what they quantify though: $\epsilon_p$ is in "units" of utility, while $\gamma_p$ is in "units" of probability.

COUP never eliminates configurations because it needs them in later phases to make statistical guarantees about relative optimality, but we know it will stop running them well before we will be able to consider them for elimination anyway. Define $\alpha_p(m, \kappa) := \sqrt{\frac{\ln(36 p^2 n_p m^2 (\log \kappa + 1)^2 / \delta)}{2m}}$ to be the confidence width, chosen to satisfy Hoeffding's inequality and the necessary union bounds. The $p$-th phase of an execution of COUP will be called *clean* if $\left| \widehat{F}_i - F_i(\kappa_i) \right| \le \alpha_p(m_i, \kappa_i)$ and $\left| \widehat{U}_i - U_i(\kappa_i) \right| \le (1 - u(\kappa_i)) \cdot \alpha_p(m_i, \kappa_i)$ for all $m_i$ and $\kappa_i$ for all configurations $i \in \mathcal{A}_p$.

**Lemma 3.** *An execution of COUP is clean for all phases with probability at least $1 - \frac{\delta}{2}$.*

**Lemma 4.** *During a clean execution of COUP we have $LCB_i \le U_i \le UCB_i$ and $UCB_i - LCB_i \le 2\alpha_p(m_i, \kappa_i) + u(\kappa_i)\left(1 - F_i(\kappa_i)\right)$ for all configurations $i \in \mathcal{A}_p$ in all phases $p$.*

The $p$-th phase of an execution of COUP will be called *characteristic* if there exists some $i_p \in \mathcal{A}_p$ with $U_{i_p} \ge OPT^{\gamma_p}$.

**Lemma 5.** *An execution of COUP is characteristic for all phases with probability at least $1 - \frac{\delta}{2}$.*

Finally, we can state our main theorem about COUP's performance.

**Theorem 2.** *If COUP is run with parameters $(\epsilon_1, \epsilon_2, ...)$ and $(\gamma_1, \gamma_2, ...)$ then with with probability at least $1 - \delta$ it returns an $(\epsilon_p, \gamma_p)$-optimal configuration at the end of every phase $p = 1, 2, ...$.*

## 5   EMPIRICAL EVALUATION

We first compare the performance of OUP with the performance of UP (Section 5.1), and with a naive procedure also considered in our previous work (Graham et al., 2023b) that simply takes

---

**Algorithm 2** COUP

---

1: **Input:** distribution over configurations $\mathcal{D}_{\mathcal{A}}$; instances $j = 1, 2, ...$; utility function $u$; failure parameter $\delta$; phase parameters $\{\epsilon_p, \gamma_p\}_{p=1,2,3,...}$.
2: $n_0 \leftarrow 0$
3: $\mathcal{A}_0 \leftarrow \emptyset$
4: **for** $p = 1, 2, 3, ...$ until terminated **do**
5:      $n_p \leftarrow \left\lceil \frac{\ln \frac{\pi^2 p^2}{3\delta}}{\gamma_p} \right\rceil$
6:      $N_p \leftarrow$ sample $n_p - |\mathcal{A}_{p-1}|$ new configurations from $\mathcal{D}_{\mathcal{A}}$
7:      $\mathcal{A}_p \leftarrow \mathcal{A}_{p-1} \cup N_p$                  ▷ all configurations for this phase
8:      **for** $i \in N_p$ **do**                  ▷ initializations for new configurations
9:          $LCB_i \leftarrow 0, UCB_i \leftarrow 1; \widehat{U}_i \leftarrow 0; \widehat{F}_i \leftarrow 0; m_i \leftarrow 0; \kappa_i \leftarrow 1$
10:      **end for**
11:      **for** $i \in \mathcal{A}_{p-1}$ **do**                  ▷ update bounds for existing configurations
12:          $UCB_i \leftarrow \widehat{U}_i + \left(1 - u(\kappa_i)\right) \cdot \alpha_p(m_i, \kappa_i)$
13:          $LCB_i \leftarrow \widehat{U}_i - \alpha_p(m_i, \kappa_i) - u(\kappa_i)(1 - \widehat{F}_i)$
14:      **end for**
15:      **while** $\max_{i \in \mathcal{A}_p} UCB_i - \max_{i \in \mathcal{A}_p} LCB_i \geq \epsilon_p$ **do**          ▷ phase termination condition
16:          $i \leftarrow \arg\max_{i' \in \mathcal{A}_p} UCB_{i'}$
17:          $m_i \leftarrow m_i + 1$
18:          **if** $2\alpha_p(m_i, \kappa_i) \leq u(\kappa_i)(1 - \widehat{F}_i)$ **then**          ▷ captime doubling condition
19:              $\kappa_i \leftarrow 2\kappa_i$
20:              $t_{i1}(\kappa_i), ..., t_{im_i}(\kappa_i) \leftarrow$ runtime of $i$ on instances $1, ..., m_i$ with timeout $\kappa_i$
21:          **else**
22:              $t_{im_i}(\kappa_i) \leftarrow$ runtime of $i$ on instance $m_i$ with timeout $\kappa_i$
23:          **end if**
24:          $\widehat{F}_i \leftarrow \frac{|\{j \in [m_i] : t_{ij}(\kappa_i) < \kappa_i\}|}{m_i}$          ▷ fraction of runs that completed
25:          $\widehat{U}_i \leftarrow \frac{1}{m_i} \sum_{j=1}^{m_i} u\left(t_{ij}(\kappa_i)\right)$          ▷ empirical average utility
26:          $UCB_i \leftarrow \widehat{U}_i + \left(1 - u(\kappa_i)\right) \cdot \alpha_p(m_i, \kappa_i)$
27:          $LCB_i \leftarrow \widehat{U}_i - \alpha_p(m_i, \kappa_i) - u(\kappa_i)(1 - \widehat{F}_i)$
28:      **end while**
29: **end for**
30: **return** $\arg\max_{i \in \mathcal{A}_p} LCB_i$

---

enough runtime samples to satisfy Hoeffding's inequality for each configuration, and takes them at a high enough captime that it can still make its guarantee whether they complete or not. We then explore the effects on performance when we expand our search to an infinitely-large set of configurations (Section 5.2). We show that the search of the configuration space does not cost COUP too much in terms of runtime, and we demonstrate COUP's flexibility in the way it performs its search. We perform our experiments on three existing datasets from the algorithm configuration literature. The minisat dataset represents the measured execution times of the SAT solver by that name on a generated set of instances (see Weisz et al. (2018) for details). The cplex_rcw and cplex_region datasets represent the predicted solve times of the CPLEX integer program solver on wildlife conservation and combinatorial auction problems, respectively (see Weisz et al. (2020) for details). In Section 5.3 we compare COUP and OUP to a variety of other procedures that do not make utilitarian optimality guarantees. We focus on the two utility functions considered in our previous work (Graham et al., 2023b). The "log-Laplace" utility function is defined as $u_{LL}(t) = 1 - \frac{1}{2}\left(\frac{t}{\kappa_0}\right)^{\alpha}$ if $t < \kappa_0$ and $u_{LL}(t) = \frac{1}{2}\left(\frac{\kappa_0}{t}\right)^{\alpha}$ otherwise. The "uniform" utility function is defined as $u_{unif}(t) = 1 - \frac{t}{\kappa_0}$ if $t < \kappa_0$ and $u_{unif}(t) = 0$ otherwise. The parameter $\kappa_0$ is set to 60 in both cases, and the parameter $\alpha$ is set to 1. To facilitate the comparison with UP we use the original doubling condition when recreating our previous experiments (Graham et al., 2023b, Figures 1 and 2). For other experiments we use the improved doubling condition. We perform all experiments in an environment where restarting runs is not possible. Code to reproduce all experiments can be found at https://github.com/drgrhm/coup.

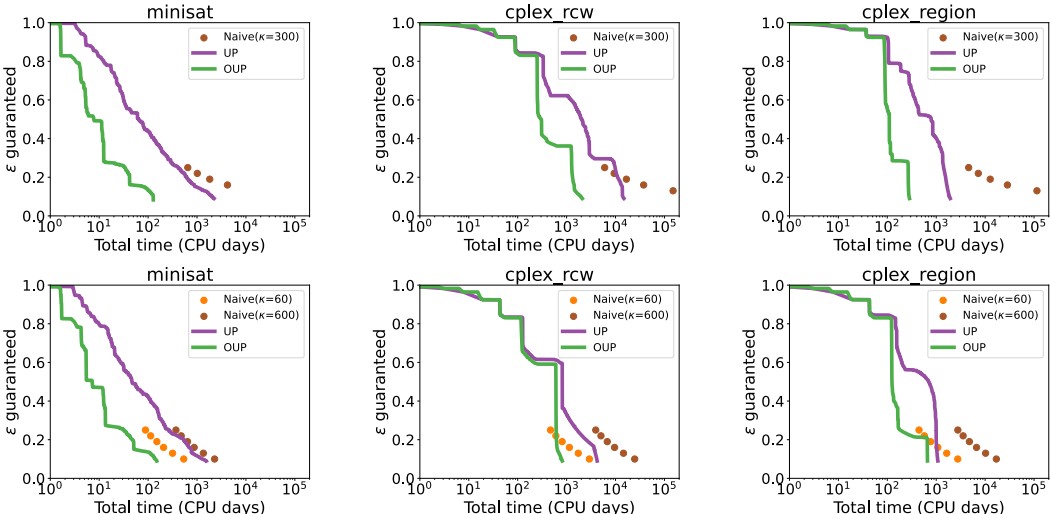

Figure 1: The $\epsilon$ guaranteed by each procedure as a function of total runtime using the log-Laplace utility function (top row) and uniform utility function (bottom row). OUP consistently outperforms both UP and the naive procedure for reasonable values of $\epsilon$, often by an order of magnitude and even when the the parameter of the naive procedure has been well-chosen.

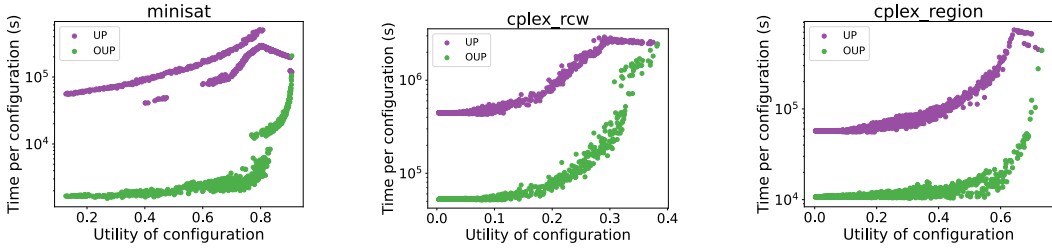

Figure 2: Total time spent by OUP on each configuration (log-Laplace utility function). Configurations are sorted according to average utility. OUP spends less time on all but the very best configurations.

## 5.1 THE CASE OF FEW CONFIGURATIONS

We first compare the performance of OUP with that of UP by recreating our previously published experiments (Graham et al., 2023b). Figure 1 shows that OUP consistently proves a meaningful $\epsilon$ in much less time than UP. OUP is faster by an order of magnitude or more (compare the top row of Figure 1 to Figure 2 in Graham et al. (2023b)). Furthermore, a very simple naive approach based on Hoeffding's inequality can be faster than UP in some scenarios (i.e. for relatively small values of $\epsilon$) when the captime parameter is chosen appropriately for the given utility function. However, OUP is consistently faster than both UP and this naive approach, regardless of the chosen captime parameter. In Section 3 we argued that OUP will spend less time running bad configurations than UP because it can stop devoting attention to them and focus on more promising ones instead. UP, on the other hand, continues to run suboptimal configurations until they are eliminated. Figure 2 shows that OUP does indeed spend considerably less time on suboptimal configurations.

## 5.2 THE CASE OF MANY CONFIGURATIONS

We now compare the performance of COUP with that of OUP to show that COUP's exploration of the configuration space does not cost too much. The results for the log-Laplace utility function are shown in Figure 3. We set $\delta = 0.01$ throughout. For each phase $p$ we set $\epsilon_p = e^{-p/6}$ and $\gamma_p = e^{-p/3}$. This schedule allows COUP to sample a large fraction of the configurations and the instances in the datasets we consider. When it is finished each phase, it can make an $\epsilon_p$-optimality guarantee with respect to the set $\mathcal{A}_p$ of $n_p$ configurations. We compare the total configuration time of COUP at

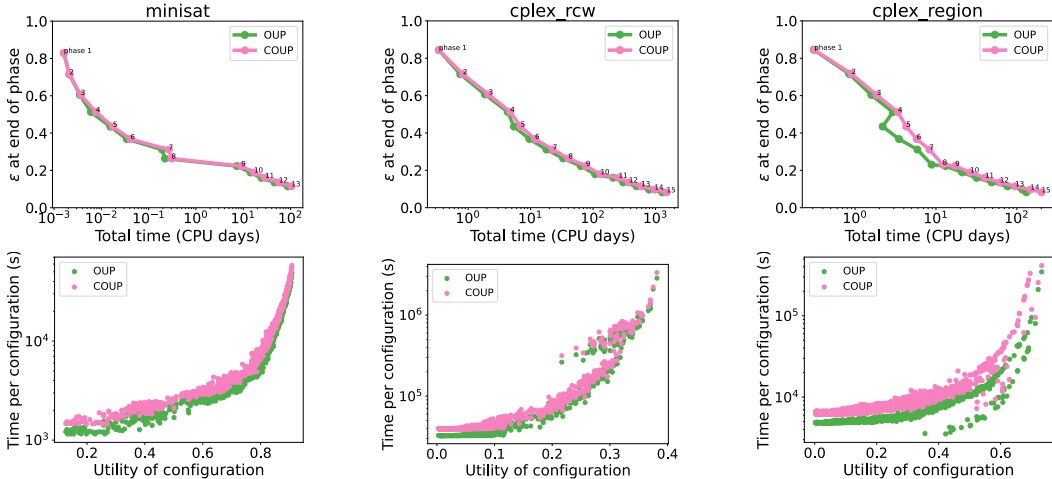

Figure 3: Performance of COUP compared to OUP, using the log-Laplace utility function. COUP is anytime, but takes only a small factor more time than OUP. The bottom row shows the final times. OUP backtracks on the `cplex_region` dataset because COUP sampled a relatively good configuration in phase 5, which OUP is able to capitalize on, proving optimality in less total time than in phase 4.

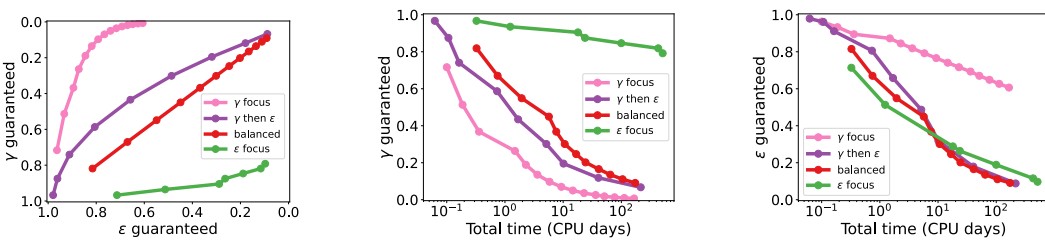

Figure 4: Different schedules for exploring new configurations (refining $\gamma$) vs. exploring existing configurations (refining $\epsilon$) when running COUP. Results are for the log-Laplace utility function and the `cplex_rcw` dataset.

the end of each phase to the time OUP takes to prove $\epsilon_p$-optimality when applied directly to this set of $n_p$ configurations. Figure 3 shows that the total configuration time required by COUP is not much different from the time required by OUP, indicating that COUP was efficient at exploring the configuration space.

COUP is extremely flexible in the way it explores new configurations relative to exploring new instances. The parameter sequences $\gamma_1, \gamma_2, ...$ and $\epsilon_1, \epsilon_2, ...$ control the relative rates at which this exploration takes place. We demonstrate this flexibility in Figure 4, presenting various different exploration strategies. The "$\gamma$ focus" strategy sets $\epsilon_p = e^{-p/30}$ and $\gamma_p = e^{-p/3}$, and so works to explore many configurations, making relatively weak guarantees about their optimality. The "$\epsilon$ focus" strategy sets $\epsilon_p = e^{-p/3}$ and $\gamma_p = e^{-p/30}$, and so explores relatively few configurations, but makes strong optimality guarantees with respect to these. The "balanced" strategy sets $\epsilon_p = \gamma_p = e^{-p/5}$. In general, it is not wise to focus on $\epsilon$ too intently until later in the search process. Once we have found a good configuration (i.e., one with a relatively large upper confidence bound), new configurations can be ruled out relatively cheaply, because they quickly look worse than the good one. So it is helpful to find a good configuration quickly, and then later work on proving that it is good. The "$\gamma$ then $\epsilon$" strategy achieves this by setting $\epsilon_p = e^{-p^3/300}$ and $\gamma_p = e^{-p^2/30}$.

## 5.3 COMPARISON TO PROCEDURES WITH DIFFERENT GUARANTEES

Finally, while we see the main contribution of COUP as being its ability to search infinite parameter spaces and make utilitarian near-optimality guarantees with respect to them, which other configuration procedures have not been able to do, some other procedures do make different but related guarantees. Comparing the performance of these various methods is not always easy since they

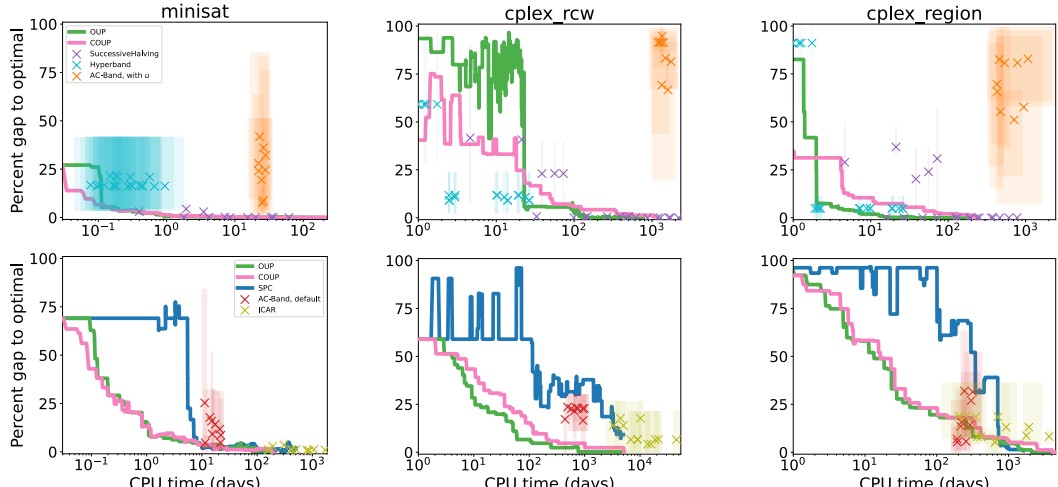

Figure 5: Average configurator performance as measured by the percentage gap to best configuration in the dataset. Top row shows procedures optimizing the log-Laplace utility function. Bottom row shows procedures optimizing an expected runtime. Error regions show maximum and minimum observations.

optimize different metrics and offer different guarantees. We compare COUP and OUP to Successive Halving (Jamieson & Talwalkar, 2016), Hyperband (Li et al., 2018), AC-Band (Brandt et al., 2023), ImpatientCapsAndRuns (ICAR) (Weisz et al., 2020) and Structured Procrastination with Confidence (SPC) (Kleinberg et al., 2019). We describe each of these procedures and their parameter settings in more detail in the appendices. Figure 5 shows the percentage gap to best configuration for each procedure at various amounts of runtime. The top row shows utilitarian procedures optimized for the log-Laplace utility function, while the bottom row shows procedures that optimize runtime. In the latter case, we optimized OUP and COUP on a uniform utility function, which is equivalent to optimizing $\kappa_0$-capped runtime. For the minisat dataset we set $\kappa_0 = 100$ and for the CPLEX datasets we set $\kappa_0 = 3000$. These values are similar to the per-configuration captimes used by ICAR and SPC, which both optimize an expected quantile-capped runtime. Figure 5 shows the results for a quantile parameter of $\delta = .1$. The results for $\delta = .2$ are essentially the same (see Appendix C). To avoid an unfair comparison we report the performance of OUP and COUP with respect to this quantile-capped runtime metric. We observed that AC-Band performed poorly with log-Laplace utility functions, so we additionally report its performance gap with respect to its default metric. We plot the averages and error regions over five seeds. OUP, COUP and SPC are anytime so they make recommendations throughout their execution, while the other procedures make point recommendations. Overall, COUP and OUP performed better than many runs of the other procedures and eventually narrowed in on the optimal configuration for every seed. Hyperband also worked well by this percentage-gap metric in some settings on some seeds. For the RCW dataset it found a good configuration more quickly than COUP but still left a gap to the best, whereas COUP and OUP continually improved. COUP and OUP avoided the "luck of the draw" that other non-anytime methods faced in the choice of their budget parameters; our investigation of varying parameterizations for these procedures shows that these hard-to-set parameters substantially impacted performance.

## 6 CONCLUSION

We have presented COUP, an improved method for utilitarian algorithm configuration. COUP shares the positive qualities of its predecessors, but is truly general purpose, in that it can search over infinite configuration spaces effectively. COUP is also shown to be superior when applied to finite configuration spaces. For the sake of continuity with previous work we have made comparisons on the datasets and utility functions that have been used before. However, this comparison is somewhat limited; a more comprehensive investigation would be a valuable direction for future work. Moreover, while we are convinced that optimizing utility functions is better than optimizing runtime, we recognize that coming up with the right utility function is not always easy for end users; establishing a tool or guide for doing this is another important future direction.

## ACKNOWLEDGMENTS

This work was funded by an NSERC Discovery Grant, a CIFAR Canada AI Research Chair (Alberta Machine Intelligence Institute), and computational resources provided both by UBC Advanced Research Computing and a Digital Research Alliance of Canada RAC Allocation.

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
