*Proof.* For any configuration $i$ and any round $r$, let $m_i(r)$ and $\kappa_i(r)$ be the number of samples taken by $i$ and the captime they were taken with, respectively, in round $r$. Let $\widehat{F}_i(m_i, \kappa_i)$ and $\widehat{U}_i(m_i, \kappa_i)$ be OUP's internal variables $\widehat{F}_i$ and $\widehat{U}_i$ after taking $m_i$ samples at captime $\kappa_i$. Define the good events:

$$
\begin{aligned}
\mathcal{G}^{(F)}_{i,m_i,\kappa_i} &= \left\{ \left| \widehat{F}_i(m_i, \kappa_i) - F_i(\kappa_i) \right| \leq \alpha(m_i, \kappa_i) \right\} \\
\mathcal{G}^{(U)}_{i,m_i,\kappa_i} &= \left\{ \left| \widehat{U}_i(m_i, \kappa_i) - U_i(\kappa_i) \right| \leq (1 - u(\kappa_i))\alpha(m_i, \kappa_i) \right\} \\
\mathcal{G}_{i,m_i,\kappa_i} &= \mathcal{G}^{(F)}_{i,m_i,\kappa_i} \cap \mathcal{G}^{(U)}_{i,m_i,\kappa_i} \\
\mathcal{G}_{i,r} &= \mathcal{G}_{i,m_i(r),\kappa_i(r)} \\
\mathcal{G}_i &= \bigcap_{r=1}^{\infty} \mathcal{G}_{i,r} \\
\mathcal{G} &= \mathcal{G}_1 \cap \cdots \cap \mathcal{G}_n.
\end{aligned}
$$

By definition, an execution is clean if and only if $\mathcal{G}$ holds. Due to the captime doubling condition, and because $i$ might eventually be eliminated (though it might not be), if $R_i$ is the set of rounds in which $i$ was selected, we have

$$
\bigcap_{m_i=1}^{\infty} \bigcap_{l_i=1}^{\infty} \mathcal{G}_{i,m_i,2^{l_i-1}} \subseteq \bigcap_{r \in R_i} \mathcal{G}_{i,m_i(r),\kappa_i(r)}. \tag{3}
$$

For any round $r$, let $\rho_i(r) \leq r$ be the last round in which configuration $i$ was selected (i.e., up to and including round $r$). Since $i$ was not selected in any rounds between $\rho_i(r)$ and $r$, the values of $i$'s internal variables in round $r$ are the same as they were in round $\rho_i(r)$. So the clean bounds hold in round $r$ if and only if they hold in round $\rho_i(r)$, and so they hold in all rounds if and only if they hold in all rounds where $i$ is selected. In other words

$$
\bigcap_{r=1}^{\infty} \mathcal{G}_{i,r} = \bigcap_{r \in R_i} \mathcal{G}_{i,r}. \tag{4}
$$

We can now bound the probability that an execution is not clean:

$$
\begin{aligned}
\Pr(\overline{\mathcal{G}}) = \Pr(\overline{\mathcal{G}_1} \cup \cdots \cup \overline{\mathcal{G}_n}) && \text{(de Morgan's law)} \\
\leq \sum_{i=1}^{n} \Pr(\overline{\mathcal{G}_i}) && \text{(union bound)} \\
= \sum_{i=1}^{n} \Pr\left( \overline{\bigcap_{r=1}^{\infty} \mathcal{G}_{i,r}} \right) && \text{(definition)} \\
= \sum_{i=1}^{n} \Pr\left( \overline{\bigcap_{r \in R_i} \mathcal{G}_{i,r}} \right) && \text{(Equation (4))} \\
= \sum_{i=1}^{n} \Pr\left( \overline{\bigcap_{r \in R_i} \mathcal{G}_{i,m_i(r),\kappa_i(r)}} \right) && \text{(definition)} \\
\leq \sum_{i=1}^{n} \Pr\left( \overline{\bigcap_{m_i=1}^{\infty} \bigcap_{l_i=1}^{\infty} \mathcal{G}_{i,m_i,2^{l_i-1}}} \right) && \text{(Equation (3))} \\
= \sum_{i=1}^{n} \Pr\left( \bigcup_{m_i=1}^{\infty} \bigcup_{l_i=1}^{\infty} \overline{\mathcal{G}_{i,m_i,2^{l_i-1}}} \right) && \text{(de Morgan's law)} \\
\leq \sum_{i=1}^{n} \sum_{m_i=1}^{\infty} \sum_{l_i=1}^{\infty} \Pr\left( \overline{\mathcal{G}_{i,m_i,2^{l_i-1}}} \right) && \text{(union bound)} \\
\leq \sum_{i=1}^{n} \sum_{m_i=1}^{\infty} \sum_{l_i=1}^{\infty} \left( \Pr\left( \overline{\mathcal{G}_{i,m_i,\kappa_i}^{(F)}} \right) + \Pr\left( \overline{\mathcal{G}_{i,m_i,\kappa_i}^{(U)}} \right) \right) && \text{(union bound)} \\
\leq \sum_{i=1}^{n} \sum_{m_i=1}^{\infty} \sum_{l_i=1}^{\infty} \frac{4\delta}{11 n m_i^2 l_i^2} && \text{(Hoeffding's inequality)} \\
\leq \delta && \text{(Basel problem)}.
\end{aligned}
$$

$\square$

**Lemma 2:** *For all $i$ at all points during a clean execution, we have*
$$LCB_i \leq U_i \leq UCB_i,$$
*and*
$$UCB_i - LCB_i \leq 2\alpha(m_i, \kappa_i) + u(\kappa_i)\big(1 - F_i(\kappa_i)\big).$$

*Proof.* We will use the fact that $U_i(\kappa) - u(\kappa)\big(1 - F_i(\kappa)\big) \leq U_i$ for any $\kappa$ by the law of total expectation, and that $U_i \leq U_i(\kappa)$ by the monotonicity of $u$. We have

$$
\begin{aligned}
LCB_i = \widehat{U}_i - \alpha(m_i, \kappa_i) - u(\kappa_i)(1 - \widehat{F}_i) && \text{(definition)} \\
\leq U_i(\kappa_i) - u(\kappa_i) \cdot \alpha(m_i, \kappa_i) - u(\kappa_i)(1 - \widehat{F}_i) && \text{(clean)} \\
\leq U_i(\kappa_i) - u(\kappa_i)\big(1 - F_i(\kappa_i)\big) && \text{(clean)} \\
\leq U_i && \text{(law of total expectation)} \\
\leq U_i(\kappa_i) && \text{(monotonic utility)} \\
\leq \widehat{U}_i + (1 - u(\kappa_i)) \cdot \alpha(m_i, \kappa_i) && \text{(clean)} \\
= UCB_i && \text{(definition)}.
\end{aligned}
$$

And

$$
\begin{aligned}
UCB_i - LCB_i = (2 - u(\kappa_i)) \cdot \alpha(m_i, \kappa_i) + u(\kappa_i)\big(1 - \widehat{F}_i\big) && \text{(definition)} \\
\leq 2\alpha(m_i, \kappa_i) + u(\kappa_i)\big(1 - F_i(\kappa_i)\big) && \text{(clean)}.
\end{aligned}
$$

$\square$

**Theorem 1:** *With probability at least $1 - \delta$, OUP eventually returns an optimal configuration and it returns an $\epsilon$-optimal configuration if it is run until $2\alpha_{i^{opt}} + u(\kappa_{i^{opt}})\big(1 - F_{i^{opt}}(\kappa_{i^{opt}})\big) \leq \epsilon$. Furthermore, for any suboptimal configuration $i$, if $m_i$ and $\kappa_i$ ever become large enough that*

$$2\alpha(m_i, \kappa_i) + u(\kappa_i)\big(1 - F_i(\kappa_i)\big) < \Delta_i$$

*then $i$ will never be run again, and $i$ will be outright eliminated once $m_i, \kappa_i, m_{i^{opt}}$ and $\kappa_{i^{opt}}$ are large enough that*

$$2\alpha(m_i, \kappa_i) + 2\alpha(m_{i^{opt}}, \kappa_{i^{opt}}) + u(\kappa_i)\big(1 - F_i(\kappa_i)\big) + u(\kappa_{i^{opt}})\big(1 - F_{i^{opt}}(\kappa_{i^{opt}})\big) < \Delta_i.$$

*Proof.* By Lemma 1, the execution is clean with probability at least $1 - \delta$. Suppose the execution is clean, so the bounds in Lemma 2 hold. For any configuration $i$ in any round $r$, suppose that $m_i$ and $\kappa_i$ are large enough that $2\alpha(m_i, \kappa_i) + u(\kappa_i)\big(1 - F_i(\kappa_i)\big) < \Delta_i$ at the beginning of round $r$ (i.e., before any runs are performed). We have

$$
\begin{aligned}
UCB_i &\leq LCB_i + 2\alpha(m_i, \kappa_i) + u(\kappa_i)\big(1 - F_i(\kappa_i)\big) && \text{(Lemma 2)} \\
&\leq U_i + 2\alpha(m_i, \kappa_i) + u(\kappa_i)\big(1 - F_i(\kappa_i)\big) && \text{(Lemma 2)} \\
&< U_i + \Delta_i && \text{(assumption)} \\
&\leq U_{i^{opt}} && \text{(definition)} \\
&\leq UCB_{i^{opt}} && \text{(Lemma 2)}
\end{aligned}
$$

so $i$ will not be selected in round $r$. And since $2\alpha(m_i, \kappa_i) + u(\kappa_i)\big(1 - F_i(\kappa_i)\big)$ does not change in any round that $i$ is not selected, $i$ will never be selected again. So if $2\alpha(m_i, \kappa_i) + u(\kappa_i)\big(1 - F_i(\kappa_i)\big) < \Delta_i$, then $i$ will never be run again.

Now, if OUP keeps selecting and running $i$, then we will have $2\alpha(m_i, \kappa_i) \to 0$ and $u(\kappa_i)\big(1 - F_i(\kappa_i)\big) \to 0$, so there will come a time where $2\alpha(m_i, \kappa_i) + u(\kappa_i)\big(1 - F_i(\kappa_i)\big) < \Delta_i$. In either case there will be some final round $r_i$ after which $i$ is never run again. Let $m_i$ and $\kappa_i$ be the procedure's internal parameters at the end of this round. Only the optimal configuration(s) will be run after round $\max_{i \text{ suboptimal}} r_i$. Define $\gamma_i = \Delta_i - 2\alpha(m_i, \kappa_i) - u(\kappa_i)\big(1 - F_i(\kappa_i)\big)$ and note that $\gamma_i > 0$. As $m_{i^{opt}} \to \infty$ and $\kappa_{i^{opt}} \to \infty$, there will eventually come a time where $2\alpha(m_{i^{opt}}, \kappa_{i^{opt}}) + u(\kappa_{i^{opt}})\big(1 - F_{i^{opt}}(\kappa_{i^{opt}})\big) < \gamma_i$ for any $i$. At this point we will have

$$
\begin{aligned}
UCB_i &\leq LCB_i + 2\alpha(m_i, \kappa_i) + u(\kappa_i)\big(1 - F_i(\kappa_i)\big) && \text{(Lemma 2)} \\
&= U_{i^{opt}} - \Delta_i + 2\alpha(m_i, \kappa_i) + u(\kappa_i)\big(1 - F_i(\kappa_i)\big) && \text{(definition)} \\
&= U_{i^{opt}} - \gamma_i && \text{(definition)} \\
&\leq UCB_{i^{opt}} - \gamma_i && \text{(Lemma 2)} \\
&\leq LCB_{i^{opt}} + 2\alpha(m_{i^{opt}}, \kappa_{i^{opt}}) + u(\kappa_{i^{opt}})\big(1 - F_{i^{opt}}(\kappa_{i^{opt}})\big) - \gamma_i && \text{(Lemma 2)} \\
&< LCB_{i^{opt}} && \text{(time reached)} \\
&\leq LCB_{i^*} && \text{(choice of } i^*)
\end{aligned}
$$

and so $i$ will be eliminated outright.

Now, let $i_r^*$ be the configuration with highest $LCB$ at the end of round $r$, chosen at Line 19, and let $i^{opt}$ be an optimal configuration. During a clean execution we will have

$$
\begin{aligned}
UCB_{i^{opt}} &\geq U_{i^{opt}} && \text{(Lemma 2)} \\
&\geq U_{i_r^*} && (i^{opt} \text{ is optimal}) \\
&\geq LCB_{i_r^*} && \text{(Lemma 2)}
\end{aligned}
$$

so $i^{opt}$ will not be eliminated. On the other hand, by the above, any suboptimal configuration $i$ will eventually be eliminated, so eventually only the optimal configuration(s) will remain.

Finally, suppose that a clean execution of OUP is run until $2\alpha_{i^{opt}} + u(\kappa_{i^{opt}})\big(1 - F_{i^{opt}}(\kappa_{i^{opt}})\big) \leq \epsilon$. Then we will have

$$
\begin{aligned}
U_{i^{opt}} - U_{i^*} &\leq UCB_{i^{opt}} - LCB_{i^*} && \text{(Lemma 2)} \\
&\leq UCB_{i^{opt}} - LCB_{i^{opt}} && \text{(choice of } i^*) \\
&\leq 2\alpha_{i^{opt}} + u(\kappa_{i^{opt}})\big(1 - F_{i^{opt}}(\kappa_{i^{opt}})\big) && \text{(Lemma 2)} \\
&\leq \epsilon && \text{(assumption)}
\end{aligned}
$$

so the returned configuration $i^*$ is $\epsilon$-optimal. □

**Lemma 3:** *An execution of COUP is clean for all phases with probability at least $1 - \frac{\delta}{2}$.*

*Proof.* For any $p, i, m_i, \kappa_i$, define the good events:

$$\mathcal{G}^{(U)}_{p,i,m_i,\kappa_i} = \left\{ \left| U_i(\kappa_i) - \widehat{U}_i(m_i, \kappa_i) \right| \leq (1 - u(\kappa_i)) \cdot \alpha_p(m_i, \kappa_i) \right\}$$

$$\mathcal{G}^{(F)}_{p,i,m_i,\kappa_i} = \left\{ \left| F_i(\kappa_i) - \widehat{F}_i(m_i, \kappa_i) \right| \leq \alpha_p(m_i, \kappa_i) \right\}$$

$$\mathcal{G}_{p,i,m_i,\kappa_i} = \mathcal{G}^{(U)}_{p,i,m_i,\kappa_i} \cap \mathcal{G}^{(F)}_{p,i,m_i,\kappa_i}$$

$$\mathcal{G}_{p,i} = \bigcap_{m_i=1}^{\infty} \bigcap_{l_i=1}^{\infty} \mathcal{G}_{p,i,m_i,2^{l_i}-1}$$

$$\mathcal{G}_p = \mathcal{G}_{p,1} \cap \cdots \cap \mathcal{G}_{p,n_p}.$$

An execution is clean in phase $p$ if $\mathcal{G}_p$ holds. The probability that an execution is not clean in any phase is

$$\Pr\left( \bigcup_{p=1}^{\infty} \overline{\mathcal{G}_p} \right) \leq \sum_{p=1}^{\infty} \sum_{i=1}^{n_p} \sum_{m_i=1}^{\infty} \sum_{l_i=1}^{\infty} \left( \Pr(\overline{\mathcal{G}^{(F)}_{p,i,m_i,2^{l_i}-1}}) + \Pr(\overline{\mathcal{G}^{(U)}_{p,i,m_i,2^{l_i}-1}}) \right) \quad \text{(union bound)}$$

$$\leq \sum_{p=1}^{\infty} \sum_{i=1}^{n_p} \sum_{m_i=1}^{\infty} \sum_{l_i=1}^{\infty} \frac{4\delta}{36 p^2 n_p m_i^2 l_i^2} \quad \text{(Hoeffding's inequality)}$$

$$\leq \frac{\delta}{2} \quad \text{(Basel problem)}.$$

$\square$

**Lemma 4:** The proof is completely analogous to that of Lemma 2 and is omitted.

**Lemma 5:** *An execution of COUP is characteristic for all phases with probability at least $1 - \frac{\delta}{2}$.*

*Proof.* For any $p$, and for any $i \in \mathcal{A}_p$, from the definition of $OPT^{\gamma_p}$, the probability that $U_i \leq OPT^{\gamma_p}$ is at most $1 - \gamma_p$. So the probability that all $n_p$ configurations have $U_i \leq OPT^{\gamma_p}$ is at most $(1 - \gamma_p)^{n_p}$, which is at most $(e^{-\gamma_p})^{n_p} = e^{-n_p \gamma_p} \leq \frac{3\delta}{\pi^2 p^2}$. Summing over $p = 1, 2, ...,$ the probability that in any phase $p$ all $n_p$ configurations have $U_i \leq OPT^{\gamma_p}$ is at most $\frac{\delta}{2}$. $\square$

**Theorem 2:** *If COUP is run with parameters $(\epsilon_1, \epsilon_2, ...)$ and $(\gamma_1, \gamma_2, ...)$ then with with probability at least $1 - \delta$ it returns an $(\epsilon_p, \gamma_p)$-optimal configuration at the end of every phase $p = 1, 2, ...$.*

*Proof.* By Lemmas 3 and 5 an execution is both clean and characteristic with probability at least $1 - \delta$. Because it is characteristic, there is some $i_p$ with $U_{i_p} \geq OPT^{\gamma_p}$. Let $i^*$ be the configuration returned at the end of phase $p$ during a clean and characteristic execution. We will have

$$U_{i^*} \geq LCB_{i^*} \quad \text{(Lemma 4)}$$
$$\geq \max_{i \in \mathcal{A}_p} UCB_i - \epsilon_p \quad \text{(phase termination condition)}$$
$$\geq UCB_{i_p} - \epsilon_p \quad \text{(maximum)}$$
$$\geq U_{i_p} - \epsilon_p \quad \text{(Lemma 4)}$$
$$\geq OPT^{\gamma_p} - \epsilon_p \quad \text{(characteristic)}.$$

$\square$

## B IMPROVED CAPTIME DOUBLING CONDITION

We discuss the change in doubling condition as applied to OUP, but the same argument holds for COUP more generally on a per-phase basis, as well as for UP. Line 9 of Algorithm 1 shows the doubling condition introduced by UP:

9: **if** $2\alpha(m_i, \kappa_i) \leq u(\kappa_i)(1 - \widehat{F}_i)$ **then**    $\triangleright$ captime doubling condition

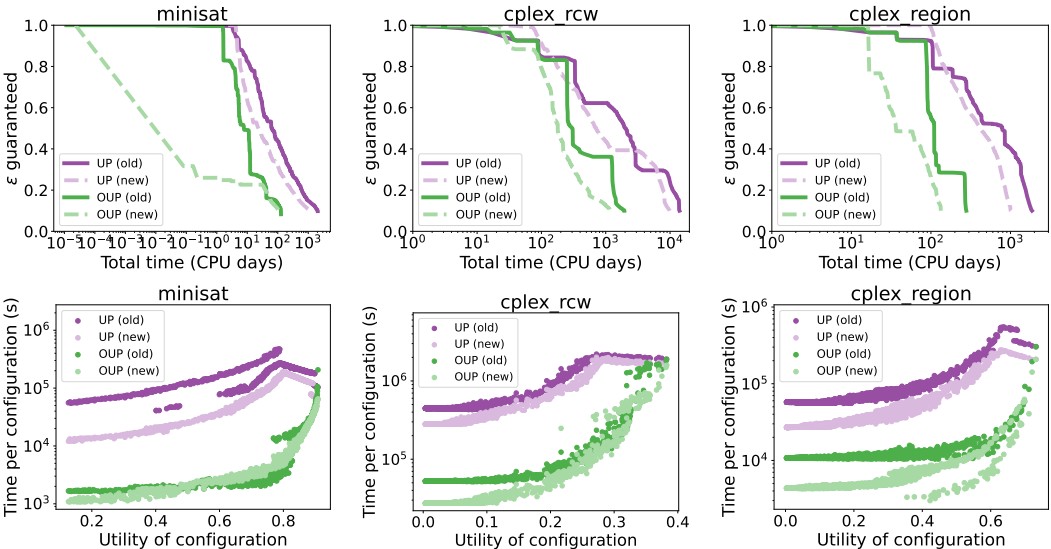

Figure 6: Improvement of the new doubling condition on UP and OUP, using the log-Laplace utility function. Other utility functions show similar improvement.

The term $2\alpha(m_i, \kappa_i)$ can be interpreted as the error incurred due to sampling because this is the term that comes from taking samples and applying the statistical bound of Hoeffding's inequality. The term $u(\kappa_i)(1 - \widehat{F}_i)$ can be interpreted as the error incurred due to capping because runs longer than $\kappa_i$ can have a utility of at most $u(\kappa_i)$, and there will only be (approximately) $1 - \widehat{F}_i$ of these. Balancing these two terms seems like a sensible approach, but a more careful analysis suggest something slightly different.

To emphasize that our uncertainty is the result of both sampling and capping, we re-write the upper and lower confidence bounds without cancelling terms as

$$UCB_i = \widehat{U}_i + \big(1 - u(\kappa_i)\big) \cdot \alpha(m_i, \kappa_i) + 0,$$
$$LCB_i = \widehat{U}_i - \big(1 - u(\kappa_i)\big) \cdot \alpha(m_i, \kappa_i) - u(\kappa_i)\big(1 - \widehat{F}_i + \alpha(m_i, \kappa_i)\big).$$

It can now be seen more clearly that the confidence width decomposes into a sum of two terms. The first represents uncertainty about runs from below the captime $\kappa_i$, and the second represents uncertainty about runs from above $\kappa_i$:

$$UCB_i - LCB_i = \underbrace{2\big(1 - u(\kappa_i)\big) \cdot \alpha(m_i, \kappa_i)}_{\text{error from runs below } \kappa_i} + \underbrace{u(\kappa_i)\big(1 - \widehat{F}_i + \alpha(m_i, \kappa_i)\big)}_{\text{error from runs above } \kappa_i}.$$

Because $i$ was chosen at line 7 to have maximum $UCB$, we have $UCB_i - LCB_i \geq UCB_{i^{opt}} - LCB_i \geq \Delta_i$, so shrinking the confidence width $UCB_i - LCB_i$ is necessary to prove a better optimality guarantee for configuration $i$. But because the confidence width is a sum of two terms, shrinking it toward zero requires shrinking both terms toward zero; the confidence width is dominated by the largest term. The new captime doubling condition we propose shrinks these two terms in a balanced way:

9′:  **if** $2\big(1 - u(\kappa_i)\big) \cdot \alpha(m_i, \kappa_i) \leq u(\kappa_i)\big(1 - \widehat{F}_i + \alpha(m_i, \kappa_i)\big)$ **then**   ▷ captime doubling condition

Empirically, this is generally an improvement over the old doubling condition and sometimes a large improvement. We set $\epsilon = \delta = .1$ and run both OUP and UP using both the old and new doubling conditions. Figure 6 clearly shows the improvement to both procedures in terms of the optimality guarantee being made and the time spent per configuration.

## C    DESCRIPTION OF PERCENTAGE-GAP EXPERIMENT

In Section 5.3 we compare COUP and OUP to the following procedures.

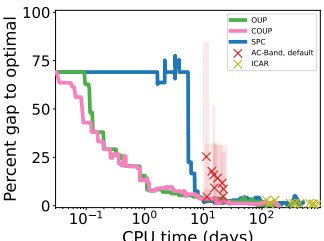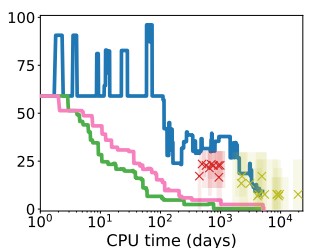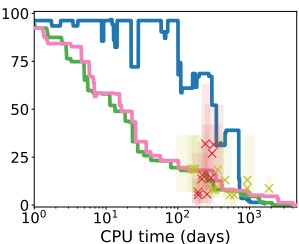

Figure 7: Average configurator performance as measured by the percentage gap to best configuration in the dataset optimizing an expected runtime. Quantile parameter $\delta = .2$

Successive Halving (Jamieson & Talwalkar, 2016) is a simple procedure that "successively halves" the number of configurations based on their performance until only one remains. It has a budget which it allocates equally to surviving configurations. We varied the budget between 22000 and 186000 for the `minisat` dataset and between 22000 and 350000 for the CPLEX datasets, which is the largest range possible based on the numbers of configurations and instances.

Hyperband (Li et al., 2018) builds on Successive Halving, allocating the budget more effectively across configurations. We set Hyperband to optimize the utility function while minimizing runtime as its resource budget. We set the multiplier $\eta \in \{2, 3, 4, 5, 6\}$ so that there would be about five brackets, as recommended by the authors, and then set the resource parameter $R$ to be as high as the dataset would permit. We ran Hyperband and Successive Halving with runs capped at $\kappa \in \{1, 10, 100, 500\}$ for the `minisat` dataset and $\kappa \in \{10, 100, 1000, 5000\}$ for the CPLEX datasets.

AC-Band Brandt et al. (2023) builds further on Hyperband, but is specifically designed for algorithm configuration. We set AC-Band to optimize the utility function, but we found this did not work very well, so we also reported its performance when optimizing its default metric, which is better. We suspect that this utility function is uniquely suited to AC-Band. AC-Band runs a number of configurations in parallel and terminates them all as soon as one finishes. Meanwhile, its default evaluation metric counts the number of times a configuration finishes first. The AC-Band paper reports the percentage gap from best average runtime, though it optimizes a different function. For consistency, we report this percentage gap. Also, we fixed what we believe to be a bug in the code from the AC-Band repo. Line 44 of the `cse.py` file calls the `env.run` function with a hard-coded captime value of 900. This is the maximum captime of the smaller, `minisat` dataset, but the larger CPLEX datasets were collected with a maximum captime of 10000 seconds. We used a captime of 10000 seconds for the CPLEX datasets (though this made only a little difference).

ImpatientCapsAndRuns (ICAR) (Weisz et al., 2020) builds on a different line of research, and is designed to discard poorly-performing configurations quickly. It measures performance according to an expected capped runtime, where the cap is set so that only a $\delta$-fraction of runs time out. Following the experiments reported in their paper, we set $\epsilon = 0.25$ for the `minisat` dataset and $\epsilon = 0.1$ for the CPLEX ones, which were the smallest values we could use without running out of instances. We used $\delta \in \{0.1, 0.2\}$ and varied $\gamma \in \{0.01, 0.02, 0.05\}$ (note that these parameters have different meaning than in our paper). We varied the boolean parameters with the same configurations as the authors.

Structured Procrastination with Confidence (SPC) (Kleinberg et al., 2019) makes the same optimality guarantee as ICAR. We ran SPC with a $\theta$-multiplier parameter of 2 and 3, and checked results after 0.1, 1, 10 and 100 CPU days of total compute time for the `minisat` dataset and 1, 10, 100 and 1000 CPU days for the CPLEX datasets. The total compute time implies a certain $\epsilon$, depending on the dataset. Since they optimize the same objective function, we used the same $\delta$ values to measure the performance of SPC as we used for ICAR.