# OpenReview forum: "Utilitarian Algorithm Configuration for Infinite Parameter Spaces"
_ICLR.cc/2025/Conference — ICLR 2025 Poster_

### Official Review · Reviewer_a4rf · 2024-11-03

**Soundness:** 3
**Presentation:** 4
**Contribution:** 2
**Rating:** 6
**Confidence:** 3

**Summary:**

The authors study the problem of utilitarian algorithm configuration with inifinite number of parameters.
The key idea is i) to adopt the UCB (upper confidence bound) method rather than the SE (sequential elimination) method in balancing exploration and confirmation of the best parameter to reach the same guarantee with shorter run time, and ii) to relax the reference point of the guarantee from the best to quantiles.

**Strengths:**

- It is well-motivated, well-executed and well-written.
- The theoretical analysis looks intuitive and right.

**Weaknesses:**

- The impact of the extension to continuous space is unclear. I would suggest to compare COUP with OUP + fixed discretization of continuous space.
- The continuous structure (similarities among neighbor parameters) is not utilized in COUP. So, C in COUP does not describe its nature accurately.

Minor:
- In Lemma 2, don’t you need “with probability 1- \delta”?
- Why not using log scale for the vertical axes of Figure 1?

**Questions:**

Can you address the points I raised in Weakness?

---

> ### Author Response · Authors · 2024-11-21
>
> "The impact of the extension to continuous space is unclear. I would suggest to compare COUP with OUP + fixed discretization of continuous space."
>
> - We see the benefit of the extension to continuous space as being the seamless applicability, in an anytime fashion, of the procedure to continuous and infinite parameter spaces, which is the reality for most algorithms. The alternative would be to sample a finite set of configurations and run OUP on this set. But how big should this set be? And what do we do if OUP finishes and we decide this set was not big enough? The only option would be to sample more configurations and then rerun OUP or, better yet, have OUP pick up where it left off, adjusting the probabilistic bounds accordingly, which is precisely what COUP does. So the real benefit of COUP over OUP is that it does this sampling internally and, importantly, does it in an anytime fashion, meaning it samples more and more configurations as time goes on. An investigation of the effect of discretization/sampling might be interesting but is not possible with our current pre-computed datasets.
>
>
> "The continuous structure (similarities among neighbor parameters) is not utilized in COUP. So, C in COUP does not describe its nature accurately."
>
> - We definitely don't want our procedure's name to be misleading. In response to your review we had a long discussion about this issue and about other candidate names. In the end, we couldn't think of a better alternative, particularly given our desire to use a name that reflects the tight connection with OUP. It seems to us that the "C" for "Continuous" does make sense because COUP can handle parameters with continuous domains (i.e., continuous in the sense of "continuous random variable") by taking an unbounded number of samples from these domains. In contrast, OUP (and UP, etc.) cannot: the set of samples they consider must be chosen in advance. We have revised the paper to make this justification for the name clearer and to forestall any potential misunderstanding.
>
>
> "In Lemma 2, don’t you need 'with probability 1 - delta'"?
>
> - Lemma 1 says that an execution is "clean" with probability $1 - \delta$, and Lemma 2 says that if the execution is clean, then the bounds hold.
>
>
> Why not using log scale for the vertical axes of Figure 1?
>
> - The updated pdf now shows total time on a log scale.

---

> > ### Comment · Reviewer_a4rf · 2024-11-22
> >
> > Thanks. I see all of my concerns have been addressed.

---

### Official Review · Reviewer_rtTp · 2024-11-04

**Soundness:** 3
**Presentation:** 3
**Contribution:** 3
**Rating:** 8
**Confidence:** 2

**Summary:**

The authors introduced a new procedure, COUP, which claims that 1) it can run on continuous space and 2) it is faster than UP in the case of discrete space. Later experiments confirmed their claims.

**Strengths:**

The paper appears organized and well-written. The authors effectively emphasized the big picture in Sections 1 and 2, providing a clear understanding of this new and improved algorithm.

The algorithm is general enough to be applied to many optimization problems. It includes theoretical properties; however, we did not check the appendix for the validity of the proof.

**Weaknesses:**

na

**Questions:**

na

---

> ### Author Response · Authors · 2024-11-21
>
> We thank the reviewer for their time and effort.

---

### Official Review · Reviewer_1WyU · 2024-11-04

**Soundness:** 4
**Presentation:** 3
**Contribution:** 3
**Rating:** 8
**Confidence:** 3

**Summary:**

The paper introduces COUP (Continuous, Optimistic Utilitarian Procrastination) procedure, which explores the parameter space - possibly uncountably infinite - of a given algorithm to optimize its performance on a specified set of inputs, with performance measured through a utility function. The finite (parameter space) verison of the procedure, known as OUP, improves upon the UP (Graham et al., 2023b) by incorporating ideas from the UCB algorithm in bandit literature. Additionally, COUP generalizes the procedure to possibly uncountably infinite parameter space. While doing so, COUP retains the theoretical guarantees of the UP algorithm while being significantly faster, as demonstrated by both theoretical guarantees and empirical results.

**Strengths:**

1) The finite version (OUP) procedure improves over the previous (UP) procedure using ideas from UCB algorithm in bandit literature.

2) Extensions to the infinite parameter space algorithms.

3) Maintaining the theoretical guarantees provided by the UP procedure.

**Weaknesses:**

1) The presentation of Section 3 could be improved slightly; introducing the notation before presenting Algorithm 1 might enhance readability.

**Questions:**

1) Anonymous repo link seems to be missing?

2) In Figure 1 - second row - third graph - there is a sudden drop in the total runtime (for OUP) around $\epsilon \approx 0.19$. Is this due to some specific structure of the problem/algorithm being analyzed?

3) The authors mention that Hyperband also performs well in the percentage-gap metric, but from Figure 5, it’s unclear to me that COUP/OUP actually outperforms it. Is there a reason to expect COUP/OUP would perform better, or is there something I might be overlooking?

---

> ### Author Response · Authors · 2024-11-21
>
> "The presentation of Section 3 could be improved slightly; introducing the notation before presenting Algorithm 1..."
>
> - We have adjusted the layout in the updated pdf so that definitions come before the algorithm description.
>
>
> "Anonymous repo link seems to be missing?"
>
> - Our intention was to release the code at time of publication; the "anonymous repo link" was just a placeholder, but we can release the code anonymously beforehand if reviewers find it important.
>
>
> "In Figure 1 - second row - third graph - there is a sudden drop in the total runtime (for OUP) around... Is this due to some specific structure of the problem/algorithm being analyzed?"
>
> - This just means that at around 700 CPU days OUP was able to prove a much better epsilon (.1) than it was able to prove before (e.g., epsilon of about .2 at 500 CPU days). This may be because OUP started seeing more instances that where more representative or it could be the result of a captime doubling event revealing something about the runtime CDFs that was not observed before.
>
>
> "... it’s unclear to me that COUP/OUP actually outperforms [Hyperband]. Is there a reason to expect COUP/OUP would perform better, or is there something I might be overlooking?"
>
> - We think Hyperband is generally a good algorithm and indeed it does beat OUP/COUP on some datasets with some parameter settings. But on others it is consistently worse (e.g., the minisat dataset). Hyperband is not anytime, so once we've committed to running it and we get an answer we cannot work to improve that answer without completely re-running it with more refined parameters. OUP/COUP work continually to improve the answer they give, until the user is satisfied, and in all cases they eventually find a configuration that is at least as good as the one found by Hyperband. Additionally, OUP/COUP make guarantees about the near-optimality of the returned configuration, which Hyperband is unable to do.

---

### Official Review · Reviewer_67Uh · 2024-11-04

**Soundness:** 3
**Presentation:** 3
**Contribution:** 3
**Rating:** 6
**Confidence:** 4

**Summary:**

The paper considers the problem of algorithm configuration. There are two key elements that are not handled in combination by previous approaches: (1) infinite (potentially continuous) parameter spaces, and (2) utilitarian reward (which is a function of the running time).

A new algorithm is proposed, COUP, which is based on UCB, using doubling trick for extending cap on the running time, and sampling increasing numbers of arms to deal with the infinite arm's space. A simplified version for finite number of arms, OUP, is also considered. The algorithms are shown to achieve close to optimal configuration.

The proposed algorithm is compared to another utilitarian algorithm UP, achieving faster convergence for the finite case. The COUP algorithm  seems to perform well empirically for the many configurations case as well.

**Strengths:**

The problem considered is important, and the proposed algorithms are sound.

The theoretical guarantees are valuable.

The paper is well written.

**Weaknesses:**

The infinite parametric space problem seems related to me to the bandit problems in continuous/metric spaces. For those problems, the performance of the bandit algorithm is compared to the optimal solution, which is facilitated by Lipschitz or stronger conditions on the reward function. In this work only the top percentile is considered (which is simpler with sampling), but I assume similar conditions on the utility could be considered here as well.

The experiments use a very limited set of baselines (UP). I think, it would have reasonable to test the algorithm against algorithms that minimize the running time (with the utility coinciding with the running time, or just using the running time as surrogate measure for those algorithms). Testing non-bandit based algorithm configuration approaches would also make sense.

**Questions:**

The questions for me are centered around the relation with metric space bandits, and the use of additional baselines.

---

> ### Author Response · Authors · 2024-11-21
>
> "... only the top percentile is considered (which is simpler with sampling), but I assume similar conditions on the utility could be considered here as well."
>
> - Stronger guarantees could indeed be made with stronger assumptions about the space of configurations. Generally in algorithm configuration, we have little idea of what structure this space has. We do not, in general, expect it to be Lipschitz continuous. Often we find that small changes to a parameter within a given range have little or no effect on performance, but if the value is pushed just outside of this range, performance changes drastically. There is a body of work showing that the parameter space is often divided into regions where performance is relatively stable, separated from each other by large jumps in performance (see, e.g., Balcan et al., (2021)), and for certain problems where this structure is known, it can indeed be taken advantage of.
>
>
> "I think, it would have reasonable to test the algorithm against algorithms that minimize the running time..."
>
> - We have now added this comparison to Figure 5 (second row) of the updated pdf. The utility function used coincides with (capped) average runtime.

---

### Official Review · Reviewer_Fgpz · 2024-11-04

**Soundness:** 4
**Presentation:** 3
**Contribution:** 4
**Rating:** 8
**Confidence:** 3

**Summary:**

An approach for algorithm configuration is presented based on a maximizing the utility of the target algorithm. The approach follows a line of work on "utilitarian procrastination (UP)", extending the previous work to function in continuous (infinite) spaces. An interesting aspect of the work is that this extension does not come at the cost of hurting performance on the finite-space case, in fact the performance improves. The resulting method, COUP, has bounds proven indicating essentially how good the configurations it finds are.

**Strengths:**

1. The paper moves the theoretical results for algorithm configuration in a very important direction, namely one step closer to real-world settings.
2. The paper is generally understandable at various levels of detail -- one need not delve into the math to understand the what and why of the paper. The math is clean and well-written, but I note that I could not completely evaluate all of the proofs.
3. The experimental results are quite good, especially in an area where the gains have been mostly small or just on different areas of the Pareto front of time vs quality. This approach is very competitive.

**Weaknesses:**

1. There are a few minor clarity issues:
	1. The start of the paper is a bit of a slog. It would have been nice to get to the point faster.
	2. I do not understand why the notation for Algorithm 1 is introduced after the algorithm is explained. I needed the notation before reading the explanation, so I ended up just being confused until I found the notation, then had to go back and read it again.
	3. In Algorithm 1, i is shadowed on line 7. Very minor, but it just seems weird (the same goes for Alg. 2).
2. I found the visuals in the experiments sometimes hard to read, and was confused by some aspects.
	1. Particularly the brown/purple/green combination has very similar shadings on some monitors/printers, another color scheme might be better.
	2. In Figure 3, the green line for OUP goes backwards between total time 10^0 and 10^1. Something must be wrong there.
	3. The symbols in Figure 5 do not match the legend, and in all honesty I can barely figure out what is going on here. The figures are just too small with too many points crowded in the same spots.

Overall, I have found no major issues with the paper, but I acknowledge that I am not an expert in the math of this paper and could have overlooked something. I am also not convinced of the superiority of the utilitarian approach of AC versus runtime configuration. However, the authors identify valid limitations and I find the use of utilitarian AC plausible for certain users, thus I do not view my disagreement on this point as something to hold against this paper.

**Questions:**

1. See issues with experiments above.
2. On line 6 of algorithm 2, COUP samples new configurations at random, which is basically how all of the theoretical approaches to AC work. Is there no way of sampling configurations that might actually be good while maintaining guarantees? It feels like these approaches are all poking around in the dark rather than actually optimizing. (future work, I suppose)

---

> ### Author Response · Authors · 2024-11-21
>
> "The start of the paper is a bit of a slog. It would have been nice to get to the point faster."
>
> - We have tried to tighten up the first two sections and we'll give particular attention to these when making our final pass.
>
>
> "notation for Algorithm 1 is introduced after the algorithm is explained"
>
> - We have now adjusted the layout so that definitions come before the algorithm and its description.
>
>
> "i is shadowed..."
>
> - Fixed, thanks.
>
>
> "brown/purple/green combination..."
>
> - We have changed Figure 1 significantly in the updated pdf. The procedures should now be more easily distinguishable based on their color and marker types.
>
>
> "In Figure 3, the green line for OUP goes backwards..."
>
> - This is to be expected sometimes and essentially has to do with the order in which the configurations are sampled. At the end of each phase COUP has proved epsilon-optimality with respect to a particular set of configurations. We give these to OUP and ask it to prove optimality for the same epsilon. What has happened here is that a good configuration was sampled in phase 5 by COUP. When we then give this set of configurations to OUP and ask it to prove epsilon-optimality, it is able to do so more quickly than it did in phase 4 because of the presence of this good configuration. We have added an explanation of this in the updated pdf.
>
>
> "The symbols in Figure 5..."
>
> - We have changed Figure 5 significantly in the updated pdf.
>
>
> "COUP samples new configurations at random... Is there no way of sampling configurations that might actually be good while maintaining guarantees?..."
>
> - Indeed, we do believe this is a very promising direction for future work. To make the theoretical guarantee we do requires independent randomly-sampled configurations. However, in addition to this, some configurations may be sampled according to a predictive model. This is the approach taken by some existing heuristic algorithm configuration procedures (e.g. SMAC). For example, half of the configurations may come from random sampling and half from the predictions of a random forest which is trained along the way. The random forest will tend to focus in on good areas of the space, and the quantile guarantee can still be made, while the total runtime increases by at most a factor of 2.

---

> > ### Comment · Reviewer_Fgpz · 2024-11-22
> >
> > Thank you for the updates. I think you have significantly enhanced the readability of the work, both at the beginning and in the experiments. I maintain my score.

---

### Official Review · Reviewer_rWFx · 2024-11-07

**Soundness:** 3
**Presentation:** 2
**Contribution:** 3
**Rating:** 5
**Confidence:** 2

**Summary:**

This paper studies utilitarian algorithm configuration, which is about automatically searching the parameter space of a given algorithm to optimise for its performance.

The first proposed algorithm is OUP.  It is a UCB style algorithm for finding configurations with good capped mean utility, with the addition of a scheme for periodically doubling the cap time.

The main algorithm is COUP. It builds on a new notion of optimisation goal called (epsilon, gamma) optimal. Epsilon is about being close to optimality and gamma controls how many configurations to sample. For fixed epsilon and gamma, the inner algorithm is for the most part just OUP. The novelty of the proposed algorithm is that it allows the sharing of trials over different epsilon and gamma.

**Strengths:**

The proposed method is general and there are extensive experiments demonstrating its performance.

**Weaknesses:**

The presentation of this paper is very hard to follow.

**Questions:**

What is delta in OUP? I don't see it being referenced in the description of OUP. Is it only used in one of the subroutines?

---

> ### Author Response · Authors · 2024-11-21
>
> "The presentation of this paper is very hard to follow."
>
> - We hope that the changes we have made help with the overall presentation.
>
>
> "What is delta in OUP?..."
>
> - Delta is the failure probability. We have specified this more clearly in OUP's inputs now.

---

### Author Response · Authors · 2024-11-21

We thank all reviewers for their time and helpful comments. We have addressed individual concerns in the comments below and we we have uploaded a new pdf of the paper that incorporates reviewers' suggestions. The notable changes are:

- Updated Figure 1 to ease readability: We have swapped the axes so that time runs left-to-right instead of bottom-to-top, which we think is more natural, and we have put total running time on a log-scale. We have also included more finely-grained data points.

- Updated Figure 5 to ease readability and make a runtime-based comparison: We have included plots for runtime-based procedures as well as utilitarian ones. We have also changed the way the averages and error regions are represented.

- Restructured text to move definitions ahead of algorithm descriptions.

---

### Meta-Review · Program_Chairs · 2024-12-17

**Metareview:**

Core technical content is bleeding onto the 11th page, for which current guidance is desk rejection. Moreover, none of the reviewers have thoroughly evaluated the main theoretical claims in terms of best arm identification with respect to the state of the art, and are overly positive while lacking any confidence about their assessments.

------

This is a revised meta-review by PCs, as the original meta-review contained a factual error which led to the reject decision.

PCs reviewed the reviews and discussions, and also consulted the manuscript. PCs concur with some of the strengths and weaknesses pointed by the reviewers. PCs recognize the interestingness of the conceptual generalization. PCs also took note of the mixed empirical success and the desiderata of having more systematic comparisons to related algorithms for this problem. Finally, PCs weigh in the factors of reviewers' confidence level in making their assessments.

The overall conclusion is that the paper is recommended to be accepted as a poster.

**Additional Comments On Reviewer Discussion:**

See above.

---

### Decision · Program_Chairs · 2025-01-22

Accept (Poster)